# Conditioning on Local Statistics for Scalable Heterogeneous Federated Learning

**Rickard Brännvall**
Department of Computer Science, RISE Research Institutes of Sweden, Luleå, Sweden
`rickard.brannvall@ri.se`

## Abstract

Federated learning is a distributed machine learning approach where multiple clients collaboratively train a model without sharing their local data, which contributes to preserving privacy. A challenge in federated learning is managing heterogeneous data distributions across clients, which can hinder model convergence and performance due to the need for the global model to generalize well across diverse local datasets. We propose to use local characteristic statistics, by which we mean some statistical properties calculated independently by each client using only their local training dataset. These statistics, such as means, covariances, and higher moments, are used to capture the characteristics of the local data distribution. They are not shared with other clients or a central node. During training, these local statistics help the model learn how to condition on the local data distribution, and during inference, they guide the client's predictions. Our experiments show that this approach allows for efficient handling of heterogeneous data across the federation, has favorable scaling compared to approaches that directly try to identify peer nodes that share distribution characteristics, and maintains privacy as no additional information needs to be communicated.

To address heterogeneous data distributions in federated learning, we propose conditioning the model on local statistics that characterize each client's joint data distribution, estimated from its own training data. Our approach relates to Personalized Federated Learning (PFL), which uses meta-learning and fine-tuning to tailor the global model to each client's data. However, PFL can be computationally intensive, lead to overfitting, and increase communication overhead Finn et al. (2017); Li et al. (2020). Clustered Federated Learning (CFL) groups clients based on similar data distributions, allowing each cluster to train a specialized model. While this improves performance, it poses challenges in determining optimal clusters and adds communication and computation overhead Sattler et al. (2019); Ghosh et al. (2020). Unlike PFL and CFL, our approach requires no modifications to the aggregation process and does not increase communication overhead.

1. *Preparation:* Each client calculates local statistics independently using their own training data. Clients have agreed on a method, but the resulting statistics are not shared.
2. *Training:* During the training phase of federated learning, each client feeds in their own local statistics as input to the model in parallel to the other training data so that the model can learn how to condition on the local data distribution. FedAvg or FedSGD can be used.
3. *Inference:* The local client uses its own static local characteristics to guide its predictions in the inference phase so that they are tailored to the specific data distribution of each client.

As many multivariate distributions are uniquely determined by their moments, we propose to use means, covariances, and higher moments to characterize the local joint distribution of features and labels. We also consider compressed statistics, e.g., by Principal Component Analysis.

Table 1: Performance comparison of conditional models with reference models on three tasks.

|  | global | cluster | client | cond |
|---|---|---|---|---|
| linreg (rmse) | 14.901 | 0.1 | 0.106 | 0.104 |
| logreg (acc) | 0.7 | 0.997 | 0.944 | 0.989 |
| emnist (acc) | 0.847 | 0.97 | 0.88 | 0.967 |

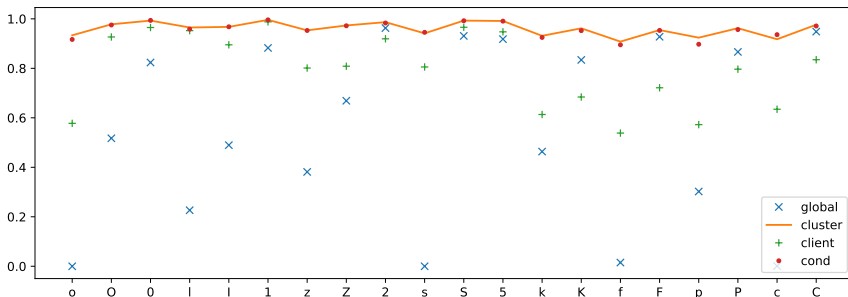

Figure 1: Conditional CNN performs better on similar characters compared to global and client unconditional reference models. Its accuracy is at par with the cluster oracle model.

**Synthetic Tasks.** We assumed a set-up of three clusters, each with 100 clients. First, we drew the true regression coefficients $\theta$ from a multivariate uniform distribution independently for each cluster. Feature vectors $x$ were drawn from a multivariate normal distribution for each client and combined with its theta according to the linear equation $y = x^T\theta + \epsilon$, adding a small Gaussian noise $\epsilon$ of 0.1 magnitude. For the logistic regression task, binary labels were obtained by thresholding at zero.

We evaluated three local conditioning models: a conditional linear model that combines features $x$ and local node stats $\mu$ by matrix multiplication; an ensemble of regression models on $x$ weighted by a softmax function that depends on $\mu_i$; and a multilayer perceptron that takes $x_j$ and $\mu_i$ as input, where $j$ index the data point and $i$ denotes the client to which it belongs. These are all global models, in the sense that all clients learn the same model with the same parameters – only $\mu_i$ differ between clients. It was calculated as the covariance between $X$ and $y$. The linear regression models were evaluated on root-mean-squared error (rmse). For Logistic Regression, we used the binary classifier equivalent of the above models trained with cross-entropy loss and evaluated on accuracy.

For comparison, we also train three conventional regressions models, $\hat{y} = x^T\beta$, where the regression weights $\beta$ were globally fitted to data from all clients, fitted separately to data from clients belonging to the same cluster, or fitted individually to each local client. We denote these as *global*, *cluster*, and *client*. Note that in a real scenario, a client wouldn't know which cluster it belongs a priori.

**EMNIST Experiment.** We also conducted experiments on the EMNIST dataset to evaluate the performance of our method on real-world data. It contains handwritten characters, including numbers, small case letters, and capital letters. To simulate heterogeneous client data distributions, we distributed the data so that each client received approximately 2500 data points from one of the three subsets (numbers, small case letters, or capital letters). We trained a three-layer convolutional neural network (CNN) that predicts the label $y$ from the image $x$. The model also takes local characteristic stats $\mu_i$, calculated for each client $i$ as the first principal component loading (eigenvector) of the flattened image concatenated with the one-hot encoding of the label, for the training data.

**Results.** Table 1 shows results for the proposed approach (using the ensemble model for the synthetic tasks). It is compared to the three reference models that fit a single model to all global data, data from peers in the same cluster, or individually for each client on its local data only. The (unconditional) global and client models underperform (except for the trivial case of linear regression from one client). The performance of the model that conditions on the local stats is not far behind the cluster set-up, which assumes oracle-knowledge of each client's peers. Figure 1 analyses the prediction performance for some characters in the EMNIST data set, revealing that the local conditioning model can better distinguish between similar-looking characters from different subsets.

**Conclusions.** Our experiments show that the proposed method effectively handles heterogeneous data distributions across clients by conditioning on local statistics. It is scalable, avoiding extensive data transfer and protocols for identifying similar data clusters. It also preserves privacy by not sharing aggregated data that could reveal local information. Future work should test these results on modern machine-learning tasks and other data modalities. Enhancements for compressing statistics, such as using latent embeddings for high-dimensional data like images, should also be considered.

# REFERENCES

Jian Chen et al. Fedpac: Personalized federated learning with feature alignment and classifier collaboration. *arXiv preprint arXiv:2306.11867*, 2023.

Chelsea Finn et al. Model-agnostic meta-learning for fast adaptation of deep networks. In *Proceedings of the 34th International Conference on Machine Learning*, pp. 1126–1135. PMLR, 2017.

Dashan Gao, Xin Yao, and Qiang Yang. A survey on heterogeneous federated learning, 2022. URL `https://arxiv.org/abs/2210.04505`.

Amrita Ghosh et al. An efficient framework for clustered federated learning. In *Proceedings of the 34th Conference on Neural Information Processing Systems*, 2020.

Chuan Li et al. Pyramidfl: A fine-grained client selection framework for efficient federated learning. In *Proc of the 28th Annual International Conf on Mobile Computing And Networking*, 2023.

Tian Li, Anit Kumar Sahu, Ameet Talwalkar, and Virginia Smith. Federated learning: Challenges, methods, and future directions. *IEEE Signal Processing Magazine*, 37(3):50–60, 2020.

H. Brendan McMahan et al. Communication-efficient learning of deep networks from decentralized data. In *Proceedings of the 20th International Conference on Artificial Intelligence and Statistics (AISTATS)*, 2017. URL `https://arxiv.org/abs/1602.05629`.

Felix Sattler, Simon Wiedemann, Klaus-Robert Müller, and Wojciech Samek. Clustered federated learning: Model-agnostic distributed multi-task optimization under privacy constraints. In *Proceedings of the 33rd Conference on Neural Information Processing Systems*, 2019.

Kai Yi et al. Fedgh: Heterogeneous federated learning with generalized global header. In *Proceedings of the 31st ACM International Conference on Multimedia*, pp. 8686–8696, 2023.

# APPENDIX - SUPPLEMENTARY MATERIAL

**Preliminaries.** Moments in statistics, such as the mean (first moment), variance (second moment), skewness (third moment), and kurtosis (fourth moment), are quantitative measures related to the shape of a distribution's probability density function. A result from multivariate statistics holds that many multivariate distributions are uniquely determined by their moments. This property aids in statistical estimation and hypothesis testing by allowing distributions to be characterized and compared based on their moments. For instance, the multivariate normal distribution is fully specified by its mean vector and covariance matrix (first and second moments). Principal Component Analysis (PCA) is related to moments, particularly the second moment (covariance), as it transforms data into a new coordinate system where the greatest variances lie on the principal components. It is achieved through eigenvalue decomposition of the covariance matrix and reduces dimensionality while preserving the essential dependence structure.

Federated learning is a distributed machine learning approach where multiple clients collaboratively train a model without sharing their local data, preserving privacy. This involves initializing a global model, performing local training on each client, and aggregating updates on a central server using techniques like federated averaging (FedAvg) McMahan et al. (2017). A significant challenge is dealing with heterogeneous data distributions across clients, leading to issues in model convergence and performance. This heterogeneity includes covariate shift, label shift, and concept shift Gao et al. (2022). Personalized Federated Learning (PFL) leverages Model-Agnostic Meta-Learning (MAML) Finn et al. (2017) to tailor the global model to each client's data. Clustered Federated Learning (CFL) groups clients into clusters based on data similarity, allowing each cluster to train a specialized model, improving performance and handling non-convex objectives Sattler et al. (2019). The Iterative Federated Clustering Algorithm (IFCA) alternately estimates cluster identities and optimizes model parameters for user clusters, demonstrating efficient convergence even with non-convex problems Ghosh et al. (2020).

Recent advancements include PyramidFL, Li et al. (2023), a client selection framework that fully exploits data and system heterogeneity within selected clients, and FedGH Yi et al. (2023), which

focuses on sharing a generalized global header and training it with local average representations. Similarly, Chen et al. (2023) introduced FedPAC, which aligns local representations to the global feature centroid.

**Conditional Linear Model**

*Outline of proof.* Assume we have $m$ clients indexed by $i$, each with training data $x_j, y_j \in D_i$, where $x_j$ is a vector of $k$ features concatenated with a 1. The conditional linear model predicts

$$\hat{y}_j = x_j^T W \mu_i \tag{1}$$

where $W$ is a weight matrix, and $\mu_i$ is a vector of characteristic statistics calculated for each client from its training data. A solution is provided by the client-by-client linear regression

$$W = I \quad \text{and} \quad \mu_i = (X_i^T X_i)^{-1} X_i^T Y_i \tag{2}$$

where we denoted by $X_i$ and $Y_i$ the concatenation of all $x_j^T$ and $y_j$ belonging to $D_i$.

To show this, we write the MSE loss

$$S = \frac{1}{2} \sum_{i=1}^m \sum_{j \in D_i} \left(x_j^T W \mu_i - y_j\right)^2 = \frac{1}{2} \sum_{i=1}^m \left(X_i W \mu_i - Y_i\right)^T \left(X_i W \mu_i - Y_i\right) \tag{3}$$

and differentiate

$$dS = \sum_{i=1}^m \left(\mu_i^T dW^T + d\mu_i^T W^T\right) \left(X_i^T X_i W \mu_i - X_i^T Y_i\right) \tag{4}$$

which doesn't have a unique solution for $dS = 0$. However, by substituting equation 2 in equation 4, we can show that such a solution indeed is known (albeit not unique). $\square$

**Synthetic Data Experiments.** We conducted experiments on synthetic data to evaluate the performance of our method in handling heterogeneous data distributions. For linear regression, the feature vectors $X$ were drawn from a multivariate normal distribution. The true regression coefficients $\theta$ were drawn from a multivariate uniform distribution $[-10, 10]$ independently for each cluster and then shared among clusters. True labels for the regression problem were obtained from $y = X^T \theta + \epsilon$, adding a small noise $\epsilon \sim N(0, 0.1)$. For logistic regression, we created binary classification data with varying class distributions, where the true labels were obtained by thresholding $X^T \theta$ at zero.

We evaluated three models that incorporate local statistics: a conditional linear model that combines features $x$ and local node stats $\mu$ by matrix multiplication (Model 1); an ensemble of regression models on $x$ weighted by a softmax function that depends on $\mu_i$ (Model 2); and a fully connected neural network (multilayer perceptron) that takes $x$ and $\mu_i$ as input (Model 3),

| *Model 1: Conditional linear* | *Model 2: Ensemble regression* | *Model 3: Multilayer perceptron* |
|---|---|---|
| | $u = \mu_i^T W_u$ | |
| $\hat{y} = x^T W \mu_i$ | $v = x^T W_v$ | $\hat{y} = \mathrm{MLP}(x, \mu_i; \eta)$ |
| | $\hat{y} = v^T \mathrm{softmax}(u)$ | |

where $i$ denotes the client that evaluates the method; and $W$, $W_u$, $W_v$, and $\eta$ are the parameters of the models that must be learned. These are all global models, in the sense that all clients learn the same model with the same parameters – only $\mu_i$ differ between clients and was calculated as the covariance between $X$ and $y$ on the (local) training data in our experiments.

For comparison, we also train three conventional regressions models, $\hat{y} = x^T \beta$, where the regression weights $\beta$ were 1) globally fitted to data from all clients, 2) fitted separately to data from clients belonging to the same cluster, or 3) fitted individually to each local client. We denote these as global, cluster, and local. Note that in a real scenario, a client wouldn't know which cluster it belongs to unless there are additional measures taken to identify its peers.

Table 2: Model comparison for synthetic tasks with 3 clusters and 10 features.

|              | global  | cluster | client | cond ens | cond mlp | cond lin |
|--------------|---------|---------|--------|----------|----------|----------|
| linreg (rmse)| 14.901  | 0.1     | 0.106  | 0.104    | 0.134    | 0.127    |
| logreg (acc) | 0.7     | 0.997   | 0.944  | 0.989    | 0.985    | 0.964    |

Table 3: Model comparison for synthetic tasks with 8 clusters and 10 features.

|              | global  | cluster | client | cond ens | cond mlp | cond lin |
|--------------|---------|---------|--------|----------|----------|----------|
| linreg (rmse)| 17.655  | 0.1     | 0.107  | 0.109    | 0.162    | 0.257    |
| logreg (acc) | 0.621   | 0.997   | 0.943  | 0.989    | 0.966    | 0.939    |

For the Logistic Regression, we used the binary classifier equivalent of the above models trained with cross-entropy loss. All training was done in PyTorch over 100 epochs using the AdamW optimizer with batch size 100, learning rate 0.001 and weight decay 0.001.

Our experiments tested different combinations in terms of the number of clusters, the number of clients in each cluster, the number of training data points per client, and the length of the feature vector. This paper reports results for set-ups with 3 and 8 clusters, respectively, with 100 peers per cluster, each having 100 data points. The results for other set-ups were very similar.

**EMNIST Experiment.**   We conducted experiments using the EMNIST dataset to assess the performance of our method on real-world data. This dataset includes handwritten characters such as numbers, lowercase letters, and uppercase letters. To mimic heterogeneous client data distributions, we distributed the data so that each client received approximately 2500 data points from one of the three subsets (numbers, small case letters, or capital letters). The test set was split accordly over the same clients. We trained a three-layer convolutional neural network (CNN) that predicts the label $y$ from the image $x$. The model also takes local characteristic stats $\mu_i$, calculated as the first principal component loading (eigenvector) of the flattened image concatenated with the one-hot encoding of the label for the training data.

We tested set-ups with models taking different number of principal components as inputs (including the case of zero). There was no significant improvement using more than one component for the conditional CNN. The reference models were instead provided with dummy input, as otherwise, the number of weights for the first layer of the models would have been different. For this we used principal components calculated on all data from all clients as well as vectors of zeros. This confirmed that there was no noticeable difference in performance attributable to the small differences in model capacity.

**More comments on the results.**   Table 2 and 3 have results for all three conditional models for set-ups with 3 and 8 clusters, respectively. Table 1 in the main text only reported results for the ensemble regression (model 2) for 3 clusters. The three first columns report the results for the reference models that each fit 1) a single model to all (global) data, 2) data from peers in the same cluster, or 3) client-by-client on local data only. The clusterwise regression perfectly fits the data, as should be expected, since the model assumes oracle knowledge of which cluster each client belongs. The unconditional global reference models clearly underperform, as should also be expected. Interestingly, while the clientwise linear regression for one client performs close to perfect (as each client has enough data), the clientwise logistic regression lags behind. The performance of the three models that condition the local stats is not far behind the clusterwise set-up, especially the ensemble model.

Table 4 lists the test set accuracy for all characters and models on the EMNIST task. The character recognition accuracy of the Conditional CNN is at par with the cluster-specific models that have oracle knowledge of peers. The global unconditional reference model underperforms, especially for similar characters that are easily confused, such as (z, Z, 2) and (i, I, 1). This is perhaps most visible in Figure 1 from the main text. For such sets, the global model will have the highest accuracy for the class with the most instances, which for EMNIST are the number of characters. The client-wise models suffer from underfitting due to a lack of data, as each client only has 2500 training images.

Table 4: Accuracies for all EMNIST characters

| character | client | cluster | cond | global |
|---|---|---|---|---|
| 0 | 0.965 | 0.993 | 0.994 | 0.824 |
| 1 | 0.988 | 0.996 | 0.996 | 0.882 |
| 2 | 0.919 | 0.987 | 0.983 | 0.963 |
| 3 | 0.953 | 0.992 | 0.989 | 0.99 |
| 4 | 0.952 | 0.99 | 0.991 | 0.977 |
| 5 | 0.948 | 0.991 | 0.991 | 0.918 |
| 6 | 0.968 | 0.994 | 0.994 | 0.975 |
| 7 | 0.941 | 0.992 | 0.992 | 0.987 |
| 8 | 0.926 | 0.991 | 0.989 | 0.974 |
| 9 | 0.94 | 0.992 | 0.989 | 0.966 |
| A | 0.736 | 0.966 | 0.956 | 0.917 |
| B | 0.563 | 0.924 | 0.903 | 0.897 |
| C | 0.835 | 0.976 | 0.971 | 0.948 |
| D | 0.61 | 0.897 | 0.881 | 0.818 |
| E | 0.591 | 0.924 | 0.908 | 0.86 |
| F | 0.721 | 0.955 | 0.953 | 0.928 |
| G | 0.603 | 0.933 | 0.923 | 0.829 |
| H | 0.687 | 0.957 | 0.939 | 0.877 |
| I | 0.895 | 0.967 | 0.968 | 0.49 |
| J | 0.7 | 0.941 | 0.918 | 0.781 |
| K | 0.684 | 0.962 | 0.952 | 0.834 |
| L | 0.805 | 0.933 | 0.938 | 0.903 |
| M | 0.873 | 0.975 | 0.967 | 0.944 |
| N | 0.841 | 0.975 | 0.971 | 0.961 |
| O | 0.927 | 0.978 | 0.975 | 0.517 |
| P | 0.797 | 0.962 | 0.957 | 0.866 |
| Q | 0.546 | 0.942 | 0.908 | 0.803 |
| R | 0.621 | 0.927 | 0.931 | 0.903 |
| S | 0.966 | 0.993 | 0.992 | 0.931 |
| T | 0.908 | 0.979 | 0.977 | 0.883 |
| U | 0.885 | 0.951 | 0.965 | 0.941 |
| V | 0.691 | 0.929 | 0.901 | 0.735 |
| W | 0.771 | 0.973 | 0.941 | 0.781 |
| X | 0.738 | 0.955 | 0.954 | 0.518 |
| Y | 0.765 | 0.958 | 0.954 | 0.728 |
| Z | 0.809 | 0.973 | 0.972 | 0.669 |
| a | 0.836 | 0.945 | 0.942 | 0.892 |
| b | 0.36 | 0.811 | 0.792 | 0.665 |
| c | 0.635 | 0.918 | 0.936 | 0.001 |
| d | 0.921 | 0.979 | 0.974 | 0.969 |
| e | 0.951 | 0.987 | 0.983 | 0.963 |
| f | 0.538 | 0.908 | 0.895 | 0.015 |
| g | 0.495 | 0.723 | 0.688 | 0.457 |
| h | 0.693 | 0.898 | 0.877 | 0.83 |
| i | 0.301 | 0.458 | 0.46 | 0.318 |
| j | 0.684 | 0.827 | 0.808 | 0.707 |
| k | 0.613 | 0.931 | 0.925 | 0.464 |
| l | 0.952 | 0.965 | 0.959 | 0.227 |
| m | 0.746 | 0.956 | 0.953 | 0.015 |
| n | 0.836 | 0.948 | 0.95 | 0.92 |
| o | 0.578 | 0.933 | 0.917 | 0.0 |
| p | 0.572 | 0.924 | 0.897 | 0.302 |
| q | 0.422 | 0.688 | 0.646 | 0.264 |
| r | 0.899 | 0.967 | 0.967 | 0.952 |
| s | 0.805 | 0.941 | 0.946 | 0.0 |
| t | 0.922 | 0.982 | 0.979 | 0.956 |
| u | 0.73 | 0.909 | 0.901 | 0.017 |
| v | 0.751 | 0.911 | 0.902 | 0.282 |
| w | 0.777 | 0.935 | 0.937 | 0.657 |
| x | 0.689 | 0.936 | 0.926 | 0.846 |
| y | 0.582 | 0.893 | 0.871 | 0.332 |
| z | 0.801 | 0.953 | 0.953 | 0.381 |

