# OpenReview forum: "Conditioning on Local Statistics for Scalable Heterogeneous Federated Learning (Tiny Paper)"
_ICLR.cc/2025/Workshop/MCDC — MCDC @ ICLR 2025_

### Official Review · Reviewer_4LN8 · 2025-02-25

**Rating:** 7
**Confidence:** 3
**Fit:** 5

**Summary:**

This paper introduces a novel approach to handling heterogeneous data distributions in federated learning (FL) by conditioning on local characteristic statistics computed independently by each client. Unlike Personalized Federated Learning (PFL) and Clustered Federated Learning (CFL), which introduce significant computational and communication overhead, this method operates without modifying the standard aggregation process or increasing data-sharing requirements. Each client calculates statistical properties such as means, covariances, and higher moments using its local dataset, which are then used during training and inference to tailor predictions to the local distribution without being shared. Experimental results on synthetic regression and classification tasks, as well as the EMNIST handwritten character dataset, demonstrate that this approach effectively improves model performance while preserving privacy. By avoiding explicit client clustering and additional meta-learning steps, it remains scalable and efficient. The results show that local statistical conditioning allows models to generalize better across diverse client distributions while matching or outperforming global and client-specific models. Future work could explore compressed representations of local statistics and extend the method to high-dimensional data modalities such as images and speech.

**Reason For Giving A Higher Score:**

Novelty and Practicality: The proposed approach to conditioning on local statistics in federated learning is a simple yet effective way to handle heterogeneous data distributions without increasing communication overhead. This makes it a valuable contribution to the field.

Privacy-Preserving and Scalable: Unlike existing methods such as Personalized Federated Learning (PFL) and Clustered Federated Learning (CFL), this approach does not require sharing additional information between clients, ensuring privacy while maintaining efficiency.

Theoretical Soundness: The use of local statistical moments (means, covariances, and PCA representations) is well-justified, leveraging fundamental statistical properties that uniquely characterize distributions.

Strong Empirical Results: The experiments on synthetic data and EMNIST demonstrate that the method achieves competitive performance while avoiding the need for explicit cluster formation, making it a promising alternative to traditional FL techniques.

**Reason For Giving A Lower Score:**

Limited Scope of Evaluation: While the experiments demonstrate effectiveness in small-scale settings, the approach has not been tested on larger and more complex datasets, such as ImageNet or real-world federated applications (e.g., medical or financial data).

Lack of Robustness Analysis: The paper does not evaluate the method's performance under adversarial settings, such as clients providing noisy or biased statistics. Assessing robustness would be crucial for real-world deployment.

Potential Computational Overhead: The paper does not discuss the computational cost of computing local statistics, particularly for high-dimensional data, which may limit the approach's applicability in resource-constrained environments.

No Dynamic Adaptation: The method assumes a fixed set of local statistics for all clients and tasks, but some form of adaptive selection based on data distribution shifts could improve generalization and performance.

**Strengths And Weaknesses:**

Privacy-Preserving Approach: The method ensures that no raw data or computed statistics are shared between clients, maintaining privacy while effectively handling data heterogeneity.

Scalability and Efficiency: Unlike Personalized Federated Learning (PFL) and Clustered Federated Learning (CFL), this approach does not require additional communication or computational overhead, making it suitable for large-scale FL applications.

Theoretical Justification: The use of local statistical moments (means, covariances, higher-order moments) is well-founded, leveraging the property that many multivariate distributions are uniquely determined by their moments.

Experimental Validation: The method is rigorously evaluated on both synthetic tasks and real-world data (EMNIST), demonstrating strong performance compared to global, clustered, and client-specific models.

Improved Generalization: By conditioning on local statistics, the proposed model generalizes better across diverse client distributions, achieving performance close to clustered models without requiring explicit cluster identification.

**Suggestions:**

Limited Scope of Evaluation: The experiments primarily focus on linear regression, logistic regression, and a small-scale image classification task (EMNIST). The approach should be tested on more complex datasets and deep learning models to validate its broader applicability.

Potential Sensitivity to Choice of Statistics: While the paper suggests using means, covariances, and PCA-based representations, it does not extensively analyze how different statistical choices impact model performance. A more thorough ablation study could provide insights into the optimal statistical representations.

Lack of Adaptive Mechanism: The method assumes that the same set of local statistics is useful across all clients and tasks. Introducing an adaptive mechanism that selects or weights statistics dynamically based on data distribution shifts could further enhance its robustness.

Computational Cost of Local Statistics: While the method reduces communication overhead, the computational cost of calculating higher-order statistics (e.g., covariance matrices, PCA components) at the client level is not discussed in detail. This could be a bottleneck in resource-constrained environments.

Absence of Robustness Analysis: The paper does not evaluate how the method performs under adversarial conditions, such as clients providing noisy or biased statistics. A robustness analysis would strengthen its practical viability in real-world FL scenarios.

---

### Official Review · Reviewer_yCfT · 2025-02-26

**Rating:** 6
**Confidence:** 3
**Fit:** 4

**Summary:**

This paper proposes an improved federated learning method to address the issue of heterogeneous data distributions across clients. The approach involves calculating local statistics (such as mean, covariance, etc.) at each client, which helps the model adapt to the local data distribution during training. During inference, these statistics assist the client in making more accurate predictions.

**Reason For Giving A Higher Score:**

The overall content of the paper is not yet sufficiently comprehensive, and the method is relatively simple, lacking a more detailed justification.

**Reason For Giving A Lower Score:**

Using statistical information to distinguish heterogeneous data distributions is intuitively feasible and can help reduce communication overhead while optimizing the computational process.

**Strengths And Weaknesses:**

## Strengths:
1. This approach simplifies communication and computation processes by leveraging statistical information to handle the heterogeneous data in federated learning without directly identifying peer nodes that share distribution characteristics.
2. The paper demonstrates the proposed method's effectiveness on both synthetic tasks and the EMNIST dataset.

## Weakness:
By using some simple statistical information for distinguishing heterogeneous data, this approach may not necessarily be effective in some complex data scenarios. Furthermore, the method's effectiveness has not been sufficiently validated in more complex real-world settings.

**Suggestions:**

The paper has not yet provided a more detailed description of the method. Additionally, the method itself is relatively simple and needs to be validated and further enhanced to demonstrate its effectiveness in more complex scenarios.

---

### Official Review · Reviewer_hmvZ · 2025-03-02

**Rating:** 7
**Confidence:** 3
**Fit:** 5

**Summary:**

The paper introduces a federated learning strategy designed to handle nodes with heterogeneous feature distributions. The approach conditions the model on statistical properties of local datasets, such as higher-order moments of variable distributions, to account for distribution shifts across nodes. The authors evaluated their approach across three tasks: a regression task and a classification task using synthetic data, and a classification task based on the EMNIST dataset. The results demonstrate that this method significantly outperforms a FL model that omits dataset statistics, and its performance is only slightly below that of a model trained on nodes with uniform data distributions.

**Reason For Giving A Higher Score:**

I am not familiar with the existing literature in this domain (i.e. including dataset statistics to train FL models). Thus, if this is the first approach leveraging dataset statistics in such a straightforward way, the paper could deserve a higher score.

**Reason For Giving A Lower Score:**

Similarly, I am unfamiliar with the novelty (or lack thereof) of the approach. Thus if this problem has already been well studied and more comprehensive papers exist, the score could be lowered.

**Strengths And Weaknesses:**

Strengths:

- They showcased their approach on different types of simple datasets.
- The idea is simple and would be easily applicable to data distributions where such statistics can be computed.

Weaknesses:

- As outlined in the suggestions, it would be nice to include some robustness of the results (e.g. including confidence intervals).

Otherwise, there are no clear weaknesses; the tiny paper opens up many opportunities for future work, starting with evaluating the approach on more diverse and realistic datasets.

**Suggestions:**

- Present the results in a more robust manner by including measures of uncertainty, such as using 5‑fold CV or computing 95% confidence intervals, to better assess the significance of the findings.
- Strengthen the related work section by discussing how dataset distribution statistics are used in FL and clarifying how existing approaches differ from the proposed method.
- I find the extremely high RMSE for the global model surprising. The authors should consider computing the RMSE values on normalized data for a more intuitive comparison across models.
- Consider revising terminology by replacing "multi-level perceptron" with "multilayer perceptron"
- Change the name of Table 4 in the appendix

---

### Decision · Program_Chairs · 2025-03-06

**Decision:**

Accept

**Comment:**

This paper proposes an improved federated learning method to address the issue of heterogeneous data distributions across clients. the topic is relevant to the works. Most of the reviewers liked the paper and recommended acceptace. We suggest the authors to incorporate the comments of the reviewers to further strengthen the paper. Overall,  we're recommend to accept this work to the workshop.